# Tailoring Lignin-Based Spherical Particles as a Support for Lipase Immobilization

Małgorzata Stanisz [ID], Karolina Bachosz [ID], Katarzyna Siwińska-Ciesielczyk [ID], Łukasz Klapiszewski [ID], Jakub Zdarta [ID] and Teofil Jesionowski *[ID]

Institute of Chemical Technology and Engineering, Faculty of Chemical Technology,
Poznan University of Technology, Berdychowo 4, PL-60965 Poznan, Poland
* Correspondence: teofil.jesionowski@put.poznan.pl; Tel.: +48-61-665-37-20

**Abstract:** Lignin-based spherical particles have recently gained popularity due to their characteristic and the usage of biopolymeric material. In this study, lignin-based spherical particles were prepared using choline chloride at different pH values, ranging from 2 to 10. Their dispersive, microstructural, and physicochemical properties were studied by a variety of techniques, including scanning electron microscopy, Fourier transform infrared spectroscopy, and zeta potential analysis. The best results were obtained for the particles prepared at pH 5 and 7, which had a spherical shape without a tendency to form aggregates and agglomerates. The lignin-based spherical particles were used for the immobilization of lipase, a model enzyme capable of catalyzing a wide range of transformations. It was shown that the highest relative activity of immobilized lipase was obtained after 24 h of immobilization at 30 °C and pH 7, using 100 mg of the support. Moreover, the immobilized lipase exhibited enhanced stability under harsh process conditions, and demonstrated high reusability, up to 87% after 10 cycles, depending on the support used. In the future, the described approach to enzyme immobilization based on lignin spheres may play a significant role in the catalytic synthesis of organic and fine chemicals, with high utility value.

**Keywords:** spherical particles; lignin; choline chloride; enzyme immobilization; lipase



## 1. Introduction

Lignin is a natural resource containing a variety of functional groups, including phenolic and alcohol hydroxyl as well as carbonyl groups [1,2]. The presence of such groups allows the efficient chemical modifications of the structure for further improvement of its physicochemical properties, including antioxidant and antibacterial activity as well as anti-ultraviolet properties [3,4]. Lignin can be used for the production of spherical particles. It has been reported that the colloidal stability and morphology depends on the type of raw material, the solvent used, the rate of addition of non-solvent, and, importantly, the pH [5,6]. Research has shown that pH has an especially important influence on the formation of lignin-based spheres, because the biopolymer contains carboxylic acid and acidic phenolic functional groups [7]. A shift in pH to higher values leads to the deprotonation of functional groups and therefore the stabilization of particles through increased surface charge. Contact with the non-solvent phase is minimized during the assembly of spherical particles, and it has been established that the formation of spheres occurs mostly via hydrogen bonding, π-stacking, and π–π and hydrophobic interactions [8]. As a result, lignin can be used in biomedicine [9–11], in bioremediation [12–14], as a biopesticide [15,16] or binder [17,18], in resins [19–21], and in cement composites [22,23]. Lignin has also been introduced as a support for enzyme immobilization, and it has been shown that the particles used in the immobilization process must be mechanically stable, water-insoluble, and safe for the environment [24–26]. Moreover, the particles should be inert and resistant to microbes, while featuring a variety of functional groups on their surface [27,28]. It is important for

the supports to be stable in a harsh environment, and to consist of particles of controlled size and different shapes to reduce diffusional limitations and to prevent inhibition of the protein [29]. It has been shown previously that the immobilization of enzymes on synthetic polymers may lead to protein instability and leakage [30]. Therefore, the use of biopolymers has come to be favored, especially the preparation of supports consisting of cellulose [31], chitin [32,33], chitosan [34–36], collagen [37], or starch [38,39]. Several different proteins have been immobilized on lignin-based spherical particles, including cellulase [40–42], tyrosinase [43,44], glucose oxidase [45,46], and lipase [47,48].

Lipase exhibits low stability, is sensitive to harsh environmental conditions, and displays activity over a narrow pH range [49]. However, the protein is capable of catalyzing esterification, hydrolysis, and aminolysis reactions, and can therefore be used in the preparation of detergents and cosmetics as well as in the food and pharmaceutical industries [25,50,51]. Hence, the development of advanced carriers for lipase is of the highest importance, and novel supports have been proposed to enhance the application potential of this enzyme. Park et al. immobilized lipase from *Candida rugosa* on cellulose/lignin microspheres, which were formed by the co-dissolution of precursors in ionic liquid followed by precipitation with the use of water. It was found that the addition of lignin leads to higher stability and activity of the immobilized protein than in the case of pristine cellulose. The resulting hydrogel was biocompatible and biodegradable, and could therefore be used in biomedical and biocatalytic applications [48]. Furthermore, the same research group prepared hydrogel microbeads consisting of cellulose and different biopolymers, including chitosan, carageen, lignin, and starch. The structure and charge of the system were strictly dependent on the biopolymer used. The obtained materials were decorated with magnetic particles to facilitate removal of the hydrogel microspheres. It was observed that magnetic hydrogel microbeads prepared from cellulose combined with alkali lignin or starch increased the efficiency of immobilization of lipase by a factor of up to 1.4 compared with model cellulose magnetic microbeads. The produced hybrid particles could be used as selective biosensors, adsorbents, and enzyme supports [52]. De França Serpa also synthesized magnetic lignin composite and used it for the immobilization of lipase B from *Candida antartica*. The biocatalyst exhibited excellent thermal stability, with a half-life of 480 min, and good magnetic properties. The obtained material was environmentally friendly and showed potential for the catalysis of various organic reactions, including the synthesis of ethyl oleate and 2-ethylhexyl oleate [47]. Sipponen et al. used cationic lignin nanospheres for the immobilization of lipase M from *Mucor javanicus* and cutinase from *Humicola insolens*. The system was mixed with sodium alginate to form beads. Both proteins retained high catalytic activity (up to 70%) when the volume ratio of water to hexane in the reaction mixture was 9:1, and therefore the prepared material may be used in the aqueous synthesis of esters [53].

Different types of lipases can be also immobilized on hydrophobic supports through interfacial activation [54,55]. Interfacial activation is a phenomenon that occurs in hydrophobic conditions in which the lid of the lipase active site is opened, facilitating catalytic action. Interfacial activation usually results in improvement of the activity of immobilized lipase. Guimarães et al. immobilized different types of lipases, including *Thermomyces lanuginosus* (TLL), *Candida rugosa* (CRL), *Rhizomucor miehei* (RML), *Candida antarctica*—forms A (CALA) and B (CALB)—as well as Eversa Transform 2.0, to support octyl agarose beads, which were treated with metallic and phosphate salts. It was highlighted that improved stability of enzymes was observed for CRL, TLL, and CALA and increased enzyme activity was noted for CALB [56]. CALA, CALB, CRL, as well as RML were also immobilized on octyl agarose beads by Arana-Peña et al. Different conditions were applied for each biocatalyst to tests its activities. It was observed that, with the use of RML, the reaction conditions have to be strictly controlled, while the immobilization of CALB and CRL remains similar with the use of different conditions [57]. Arana-Peña et al. co-immobilized different types of lipases, such as TLL, RML, CALA, CALB, and phospholipase Lecitase Ultra (LEU). Enzymes were immobilized through the layer-by-layer technique and it was

noted that the enzyme order, reaction conditions, and used substrate are important and influence the activity of biocatalysts [58]. Moreover, the same scientific team combined interfacial activation with ion exchange. It was observed that, enzymes with higher activity, such as CALA, TLL, and CALB, have to be immobilized first to obtain increased activity of the prepared systems [59].

In this work, lignin-based spherical particles were prepared with the use of choline chloride. The materials were fabricated at different pH values, and the samples with the most suitable dispersive-microstructural characteristics were selected for immobilization of lipase. The prepared supports are novel materials; therefore, the catalytic activity of the enzyme was tested at different values of several parameters, including the amount of support, pH, temperature, concentration of protein, and number of reuses. To our best knowledge, this is the first report on the fabrication of lignin-based spherical particles with the use of choline chloride only. Thus, we present here an innovative modification of lignin with its simultaneous application in the development of a biocatalytic system with lipase B from *Candida antarctica*.

## 2. Results and Discussion

### 2.1. Influence of pH on Formation of Spherical Particles

Lignin was combined with choline chloride at different pH values to determine which value led to a product with the most favorable properties: spherically shaped particles, a homogeneous structure, and a lack of tendency to form aggregates or agglomerates. The process was carried out at pH values of 3, 5, 7, and 10, and the dispersive and microstructural properties of the products were determined. SEM images of all samples are presented in Figure 1, and their particle size distribution and polydispersity indices are shown in Table 1. It is seen that sample G1, prepared at pH 5, exhibited the most homogeneous structure of spherically shaped particles (255–1484 nm) and had no tendency to form aggregates or agglomerates. The results for sample G2, prepared at pH 7, are similar, but the synthesized particles are larger (342–1990 nm), with a spherical but in some cases also ellipsoidal shape. The samples prepared in reactions at pH 5 and 7 also gave the best results in terms of particle size distributions with low polydispersity indices, below 0.100, which also indirectly confirms the spherical shape of the prepared material. On the other hand, samples G0 and G3, prepared at pH 3 and 10, respectively, were more irregular in shape, especially G3. It may be concluded that basic reaction conditions are less favorable for the preparation of spheres than acidic conditions. Both materials prepared in harsher environmental conditions show a tendency to form aggregates and agglomerates. It has been previously highlighted that larger particles may be obtained at lower pH because of agglomeration of the material. The ionization of carbonyl groups occurs with increasing pH, resulting in an increase in repulsive forces and the formation of particles with smaller diameters. Phenolic hydroxyl and carbonyl groups can be easily deprotonated and ionized around pH 9, which results in disintegration of the spheres [60]. Similar results were obtained by Azimvand et al., who showed that with lowering of the pH of the synthesis environment, the obtained particles tend to coagulate and agglomerate [61]. The differences in shape observed in this study may be related to partial depolymerization and dissolution of lignin in strong acidic and basic reaction conditions. Moreover, for both materials G0 and G3, the polydispersity index was 1.000, which also confirms that the particles of these samples are irregular in shape. Yan et al. prepared three different types of lignin-based spherical particles at pH values varying from 2 to 6. Different nanospheres were synthesized at different pH values, from hollow spheres to solid structures. All prepared particles were found to have a controllable size and structure, with excellent dispersion ability and long-term stability [62]. Ma et al. showed that lignin-based spherical particles are larger in size below pH 3 or above pH 10 [63]. Guo et al. also reported that at pH 9 a deprotonation process of the hydroxyl and carbonyl groups may occur, resulting in the disassembly of lignin-based spherical particles [60].

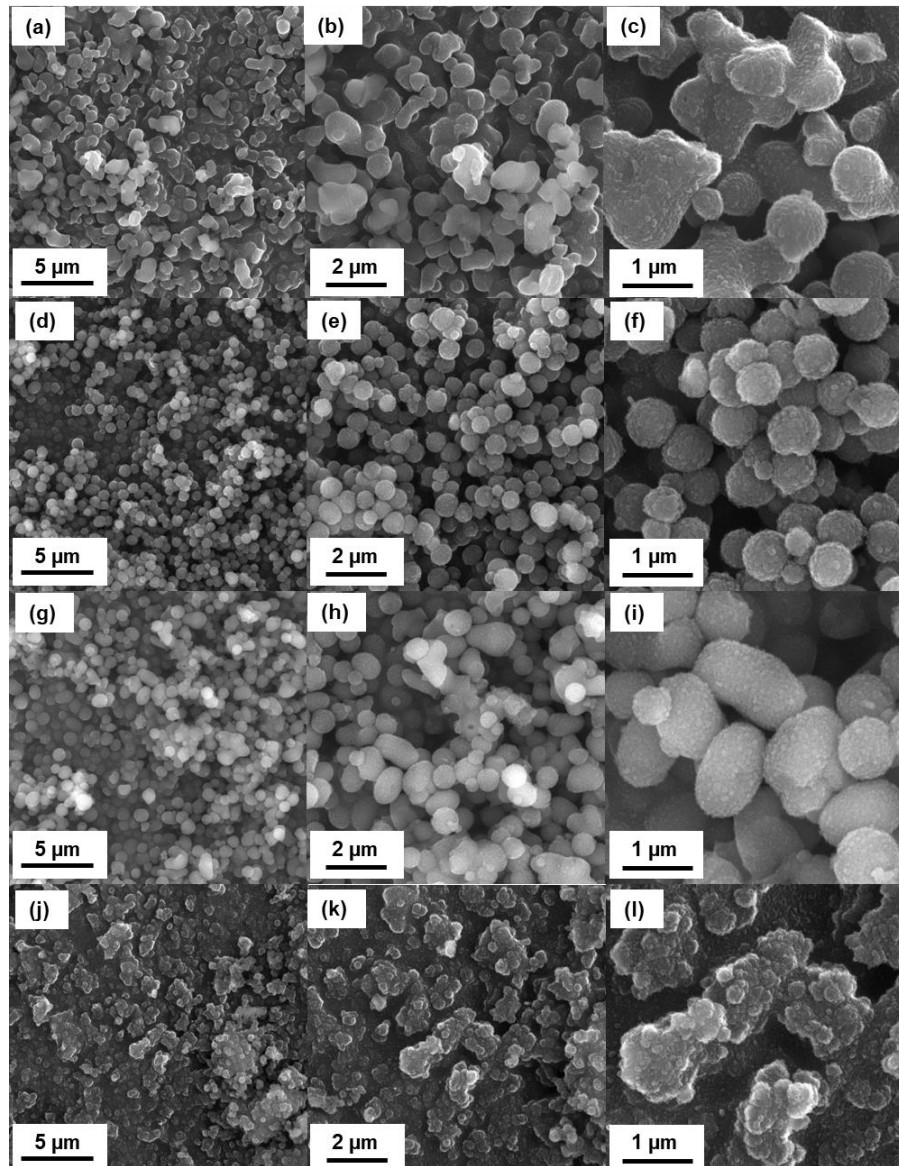

**Figure 1.** SEM images of (**a**–**c**) G0, (**d**–**f**) G1, (**g**–**i**) G2, and (**j**–**l**) G3, shown at three different magnifications.

**Table 1.** Particle size distribution and polydispersity index of the prepared samples.

| Sample | Particle Size Distribution (nm) | Polydispersity Index (PdI) |
|---|---|---|
| G0 | 459–1718 | 1.000 |
| G1 | 255–1484 | 0.091 |
| G2 | 342–1990 | 0.064 |
| G3 | 91–712; 3580–6439 | 1.000 |

This is in agreement with the results obtained in the present study, as samples G0 and G3, synthesized at pH 3 and 10, respectively, contained larger particles than samples G1 and G2. To our best knowledge, there have been no previous studies of the fabrication of lignin-based spherical particles using choline chloride only. For further study of the immobilization process, only materials G1 and G2 were selected. This is because both materials exhibited spherical shapes and no significant tendency to form aggregates or agglomerates; this means that an effective immobilization process can be performed and the diffusion resistance during subsequent catalytic reactions can be limited.

### 2.2. Physicochemical Evaluation of the Lignin-Based Spherical Matrix and Materials following Immobilization

#### 2.2.1. ATR–FTIR Spectroscopy

Characteristic bands of kraft lignin, choline chloride, free lipase, and all the prepared products were identified with the use of ATR–FTIR spectra. First, spectra were obtained to evaluate the influence of kraft lignin and choline chloride on the preparation of lignin-based spherical particles; then the analysis was performed for materials after the immobilization process to examine its effectiveness (see Figure 2).

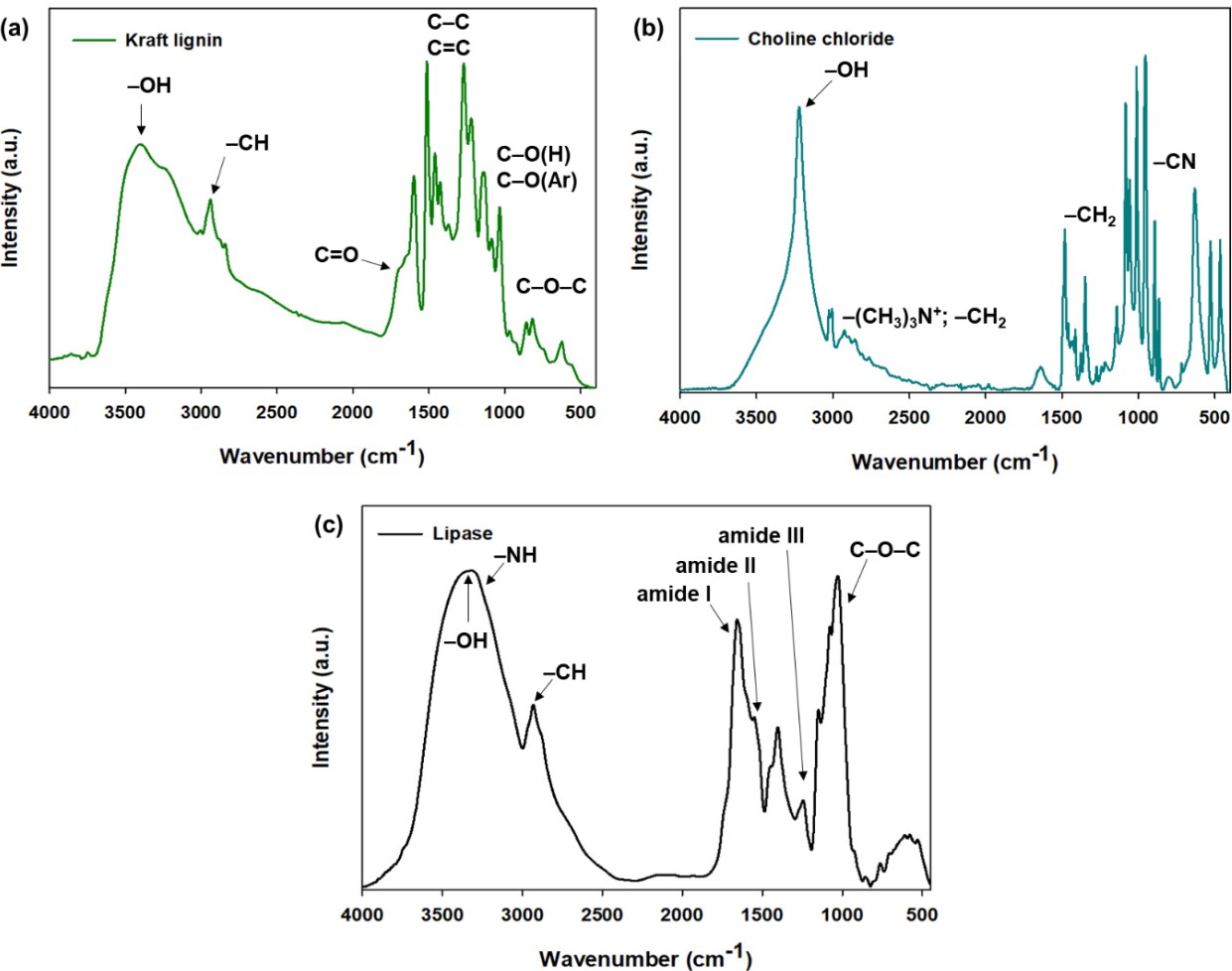

**Figure 2.** ATR–FTIR spectra of (**a**) kraft lignin, (**b**) choline chloride, and (**c**) *Candida antarctica* lipase B.

The FTIR spectrum of kraft lignin is shown in Figure 2a. The biopolymer exhibits a wide band with a maximum at 3450 cm$^{-1}$, which corresponds to the stretching vibrations of hydroxyl groups. The shape of the peak is broad, which may indicate that the sample has a higher ratio of alcoholic hydroxyl groups than phenolic ones. This is very favorable for the preparation of lignin-based spherical particles, because a higher content of phenolic hydroxyl groups may enhance the hydrogen bonding between lignin and water and therefore prevent the formation of lignin-based spherical particles [64]. Moreover, there is a sharp peak around 3000 cm$^{-1}$ and a wider peak at 2800 cm$^{-1}$, which correspond to the and aliphatic –CH groups. A small band at 1700 cm$^{-1}$ is attributed to stretching vibrations of C=O, indicating that the untreated kraft lignin has a limited amount of carbonyl functional groups in its structure. There is also a presence of characteristic functional groups associated mainly with aromatic rings, including C–C and C=C (1450 cm$^{-1}$ to 1250 cm$^{-1}$), C–O(H) and C–O(Ar) (1100 cm$^{-1}$), and C–O–C (980 cm$^{-1}$) [13,65].

The ATR spectrum of choline chloride (see Figure 2b) reflects all the characteristic functional groups related to the analyzed compound. The peak around 3250 cm$^{-1}$ can be attributed to stretching vibrations of the –OH groups. The characteristic wide band from 3050 cm$^{-1}$ to 2800 cm$^{-1}$ corresponds to stretching vibrations of the $(CH_3)_3N^+$ groups. There are also characteristic bands for –CH$_2$ and –CN around 1450 cm$^{-1}$ and 940 cm$^{-1}$, respectively [66,67].

The FTIR spectrum in Figure 2c shows the characteristic bands related to the structure of the protein. There is a broad band from 3600 cm$^{-1}$ to 3100 cm$^{-1}$ attributed to stretching vibrations of the hydroxyl and amine groups. A band attributed to –CH groups has a maximum at the wavenumber 2900 cm$^{-1}$. Characteristic bands with maxima at 1660 cm$^{-1}$, 1526 cm$^{-1}$, and 1295 cm$^{-1}$ confirm the presence of amide I, II, and III groups, respectively, in the enzyme structure. The sharp peak around 1000 cm$^{-1}$ is related to the C–O–C group [68].

FTIR spectra of materials G1 and G2 before and after the immobilization process are presented in Figure 3. It is observed that both materials exhibit similar functional groups, mostly originating from kraft lignin, independently of the pH used for their synthesis. This may indicate that the pH used for the synthesis of lignin-based spherical particles has limited influence on the presence of functional groups, mainly affecting the morphological properties of the spheres. The peaks occurring at wavenumbers 3400 cm$^{-1}$ and 3238 cm$^{-1}$ correspond to the stretching vibrations of hydroxyl groups and amino groups, respectively. The sharp band at 2950 cm$^{-1}$ corresponds to stretching vibrations of –CH groups. There is also a sharp and well-defined peak around 1710 cm$^{-1}$, which originates from the vibrations of the carbonyl groups. This is the main difference from the kraft lignin spectrum, as there is a limited amount of carbonyl groups in the pristine biopolymer structure, whereas in the spectra of the resulting spheres, the intensity of this signal is increased. The presence of C=O functional groups may also be related to the partial fragmentation and depolymerization of kraft lignin during the formation of lignin-based spherical particles. There are also bands that can be attributed to aromatic ring vibrations, at around 1450–1350 cm$^{-1}$ and 1200–1000 cm$^{-1}$ for C–O(H) and C–O(Ar).

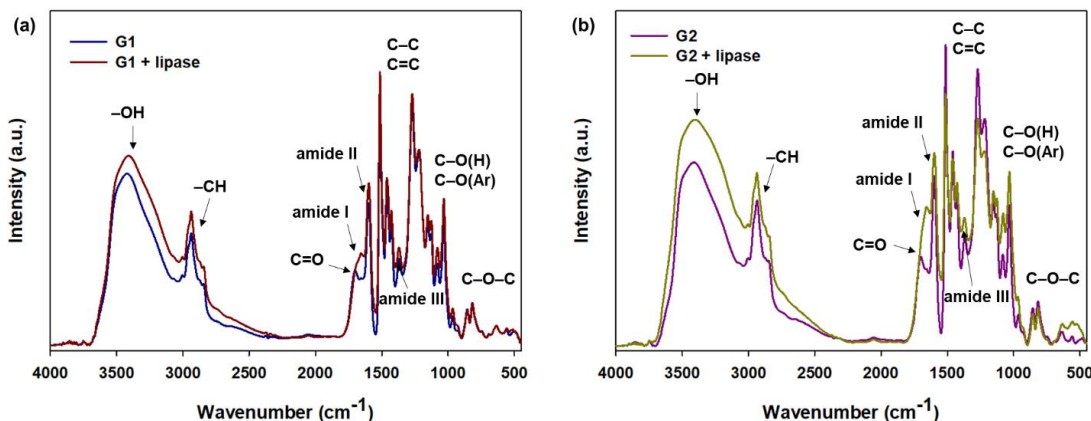

**Figure 3.** FTIR spectra of materials G1 (**a**) and G2 (**b**) before and after lipase immobilization.

The presence of a variety of functional groups, especially hydroxyl and carbonyl, can lead to the formation of interactions between the lignin-based support and the enzyme; therefore, FTIR spectra were also prepared for the immobilized material. Figure 3 confirms that there are differences in the intensity and presence of functional groups after enzyme deposition. The increased intensity in peaks of the –OH and –CH groups, as well as the presence of a new band at 1640 cm$^{-1}$, corresponding to stretching vibrations of the amide I functional groups, indicate efficient enzyme immobilization. Moreover, the amide II bands are shifted towards higher values compared with the FTIR spectrum of lipase. There is also a visible band at 1256 cm$^{-1}$, which corresponds to amide III groups. Thus, the immobilization of lipase on the surface of the lignin-based spherical particles can be confirmed, because of the presence of the characteristic functional groups originating from both the structure of the support and the protein [69].

### 2.2.2. Zeta Potential

Electrokinetic characterization was performed for pristine kraft lignin, both types of lignin-based spherical particles, and the immobilized products. The curves obtained are presented in Figure 4. All prepared samples were compared with pristine kraft lignin, which had negative zeta potential values over the whole analyzed pH range from 2 to 10, attained electrokinetic stability above pH 6, and exhibited no isoelectric point. Lipase B from *Candida antarctica* has an isoelectric point around pH 5 and exhibits a negative charge at a pH above 5 [70,71]. All prepared samples displayed excellent dispersion stability in an alkaline medium because of the ionization of the functional groups; in an acidic medium, the negatively charged functional groups are neutralized, and the system is more prone to destabilization. Samples G1 and G2 were found to have negative zeta potential values over the whole analyzed pH range. Both materials demonstrated electrokinetic stability beyond pH 2, as the samples reached a minimum zeta potential of –30 mV without an isoelectric point. Sample G1 reached the lowest value, around –65 mV at pH 10, and it can be concluded that this material has a higher content of functional groups available for ionization than G2, for which the lowest zeta potential was approximately –55 mV. On the other hand, particles may exhibit a high content of protonatable groups, including hydroxyl, carboxylic, and carbonyl, leading to a decrease in zeta potential with increasing pH. Similar results were obtained by Yan et al., who reported that, with an increase in the pH value, the negative charge of the prepared lignin-based nanoparticles also increased [62]. Moreover, as has been noted previously, samples G1 and G2 exhibited a low tendency to form aggregates and agglomerates, which may also be connected to the highly negative zeta potential values obtained for both materials, enabling the production of sufficient dual electric repulsion between particles [61,63].

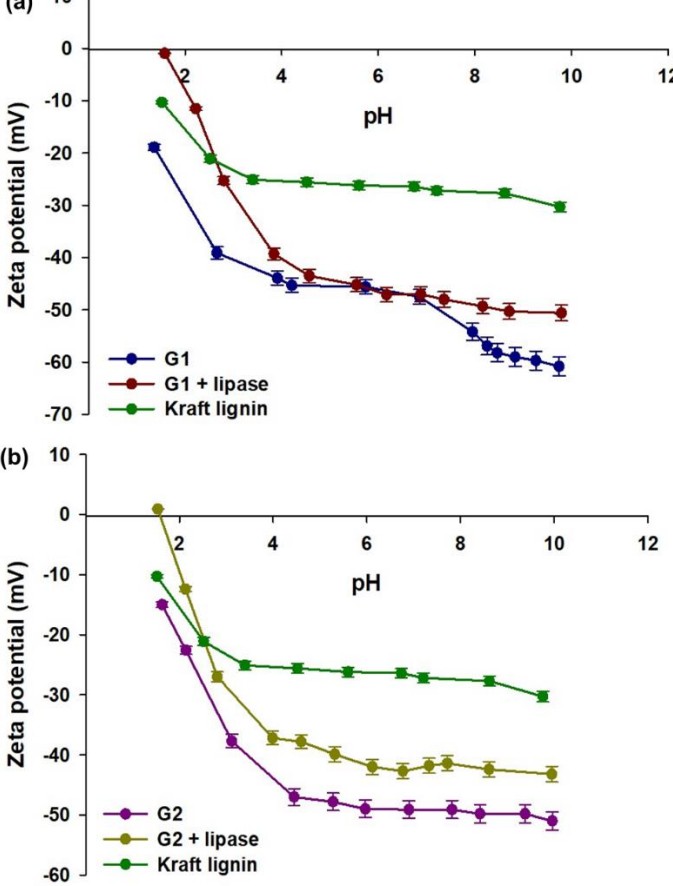

**Figure 4.** Zeta potential vs. pH presented for (**a**) G1 and G1 + lipase and (**b**) G2 and G2 + lipase, compared with kraft lignin.

Zeta potential curves were also obtained for the immobilized materials and compared with those of the pristine materials G1 and G2. Because the addition of lipase generated changes in the values of zeta potential, it can be indirectly concluded that the immobilization process was successful. Both samples following immobilization showed a shift towards higher zeta potential values compared with the support, and had an isoelectric point around pH 1.5. Therefore, it is suggested that electrostatic interactions play a key role in the deposition of enzymes on the surface of lignin-based spherical particles. Amino acid radicals in lipase and carbonyl and hydroxyl groups in lignin can undergo a dissociation process in an aqueous environment with a change in surface charges, enabling electrostatic interactions to occur [72]. The lowest zeta potential values were around –50 mV and –45 mV at pH 10, respectively, for materials G1 and G2 with immobilized lipase. It was concluded that the immobilization process was successful for both prepared materials.

### 2.2.3. Porous Structure Analysis

The BET surface area, total pore volume, and mean pore diameter were calculated based on the results of low-temperature nitrogen adsorption/desorption for samples G1 and G2 before and after the immobilization process, as well as for pristine kraft lignin (see Table 2). It was observed that the lignin-based spherical particles had a higher surface area than the pristine biopolymer. The value of this parameter increased from 4 $m^2/g$ for kraft lignin to 17 $m^2/g$ for sample G1 and 12 $m^2/g$ for G2. Kraft lignin has a large mean pore diameter (12.1 nm), and the obtained particles were characterized by smaller values of this parameter: 2.13 nm and 2.01 nm for G1 and G2, respectively. It may be concluded that the preparation of spherical particles at pH 5 results in the formation of particles with a larger surface area, higher total pore volume, and larger pores than those prepared at pH 7, suggesting better sorption ability. The structural properties of the material were also examined after the immobilization procedure, and it was found that the procedure caused changes in the porous structures of both materials. The samples with immobilized enzyme exhibited decreased values of surface area, total pore volume, and mean pore diameter, compared with the material before the immobilization process, indicating that there was effective deposition of the enzyme onto the lignin-based spherical particles and partially in their pores [73,74]. Moreover, it may be noted that material G1, because of its structural properties, displayed a higher ability to immobilize lipase than material G2.

**Table 2.** Porous structure properties of the analyzed samples.

| Sample | Structural Properties | | |
|---|---|---|---|
| | $A_{BET}$ (m$^2$/g) | $Vp$ (cm$^3$/g) | $Sp$ (nm) |
| G1 | 17 | 0.009 | 2.13 |
| G2 | 12 | 0.004 | 2.01 |
| G1 + lipase | 5 | 0.001 | 1.95 |
| G2 + lipase | 8 | 0.001 | 1.94 |
| Kraft lignin | 4 | 0.002 | 12.1 |

$A_{BET}$—the BET surface area; $V_p$—total volume of pores; $S_p$—mean size of pores.

### 2.3. Catalytic Properties of the Produced Biocatalytic Systems

To confirm the effectiveness of the lipase immobilization, both prepared biocatalytic systems were characterized in terms of the amount of immobilized enzymes as well as process yields (see Table 3).

It is seen that after 24 h of lipase immobilization, with the use of a 5 mg/mL enzyme solution, high immobilization yields were achieved for both materials (96% and 97% for G1 and G2, respectively). In addition, a relatively large amount of lipase was deposited on the lignin-based spherical particles: approximately 0.24 mg on 1 mg of both supports. Moreover, the activity recovery of lipase immobilized on the G1 and G2 supports was 92% and 85%, respectively, indicating high enzyme activity, limited interference in the enzyme structure, and low diffusional limitations. On the basis of these results, it can be concluded

that the immobilization process was effective, and that the produced biocatalytic systems have the potential to be used in hydrolysis processes.

**Table 3.** Characteristics of the enzyme before and after immobilization. The conditions of reaction were as follows: 30 °C, pH = 7, time duration 24 h, volume of a process 5 mL, 100 mg of support, and a 5 mg/mL concentration of lipase. All data are presented as the means $\pm$ standard deviation of three experiments.

| Analyzed Parameter | Free Enzyme | G1 | G2 |
|---|---|---|---|
| Immobilization yield (%) | - | $96 \pm 2.89$ | $97 \pm 2.92$ |
| Amount of immobilized enzyme (mg/mg) | - | $0.241 \pm 0.0072$ | $0.243 \pm 0.0073$ |
| Activity recovery (%) | 100 | $92 \pm 2.76$ | $85 \pm 2.57$ |

The hydrolysis reaction of *p*-nitrophenyl palmitate (*p*-NPP) was performed to determine the catalytic activity of native lipase as well as the enzyme immobilized on G1 and G2 supports. Different conditions of immobilization, including time, concentration of lipase, and amount of support, were tested to select the most suitable conditions, facilitating the obtaining of systems with high hydrolytic activity (see Figure 5). It was observed that an amount of 100 mg of the support enables the highest catalytic activity of lipase, independently of the type of lignin-based spherical particles. The immobilization process on material G2, over the whole analyzed range, enabled higher lipase activity than when the corresponding amount of the G1 support was used. Interestingly, Bracco et al. selected 100 mg of chitosan microspheres as an appropriate amount of support for the immobilization of lipase, and observed that after enzyme adsorption, the catalytic activity of the protein was high (up to 100%) [75]. A significant decrease in protein activity is observed with an increase in the amount of both materials. The use of 200 mg of support decreases the enzyme activity from 100% to 76% and 60% for materials G1 and G2, respectively. In the case of G1, the enzyme activity decreased with the use of a smaller amount of support. However, when 200 mg was used for the G1 material, the activity of lipase was higher than with material G2. This is due to the fact that the use of a large amount of support in the immobilization process can lead to a decrease in the interaction between the enzyme and specific active groups present on the lignin-based spherical particles [76].

The effect of lipase concentration on the relative activity of the immobilized protein was also evaluated, and the results are presented in Figure 5b. Five different lipase concentrations, from 0.5 mg/mL to 10 mg/mL, were used for this procedure. It was found that the highest catalytic activity could be obtained with the use of a 5 mg/mL lipase solution. Similar results were obtained by Lu et al., who reported that the best saturation of lipase on an $SiO_2$–$Fe_3O_4$–polyacrylonitrile-based support was around 4 mg/mL, with a relative activity of lipase around 90% at pH 7 [77]. On the other hand, dos Santos et al. showed that activated chitosan particles can immobilize 20 mg/g of lipase; however, the addition of a smaller amount of protein enables a satisfactory 95% immobilization yield [78]. In this study, the lowest activity of the enzyme was observed for a 0.5 mg/mL protein solution, especially for the G2 support. The low activity might also be related to the fact that from the initial enzyme solution with the lowest concentration, an insufficient amount of the enzyme was deposited to generate high activity. Moreover, independently of the enzyme concentration used, it was observed that the protein immobilized on material G1 had a higher relative activity than that deposited on the G2 support. This may be related to the more hydrophobic character of the G2 support, leading to greater disruption to the enzyme structure and reducing the activity of the resulting system.

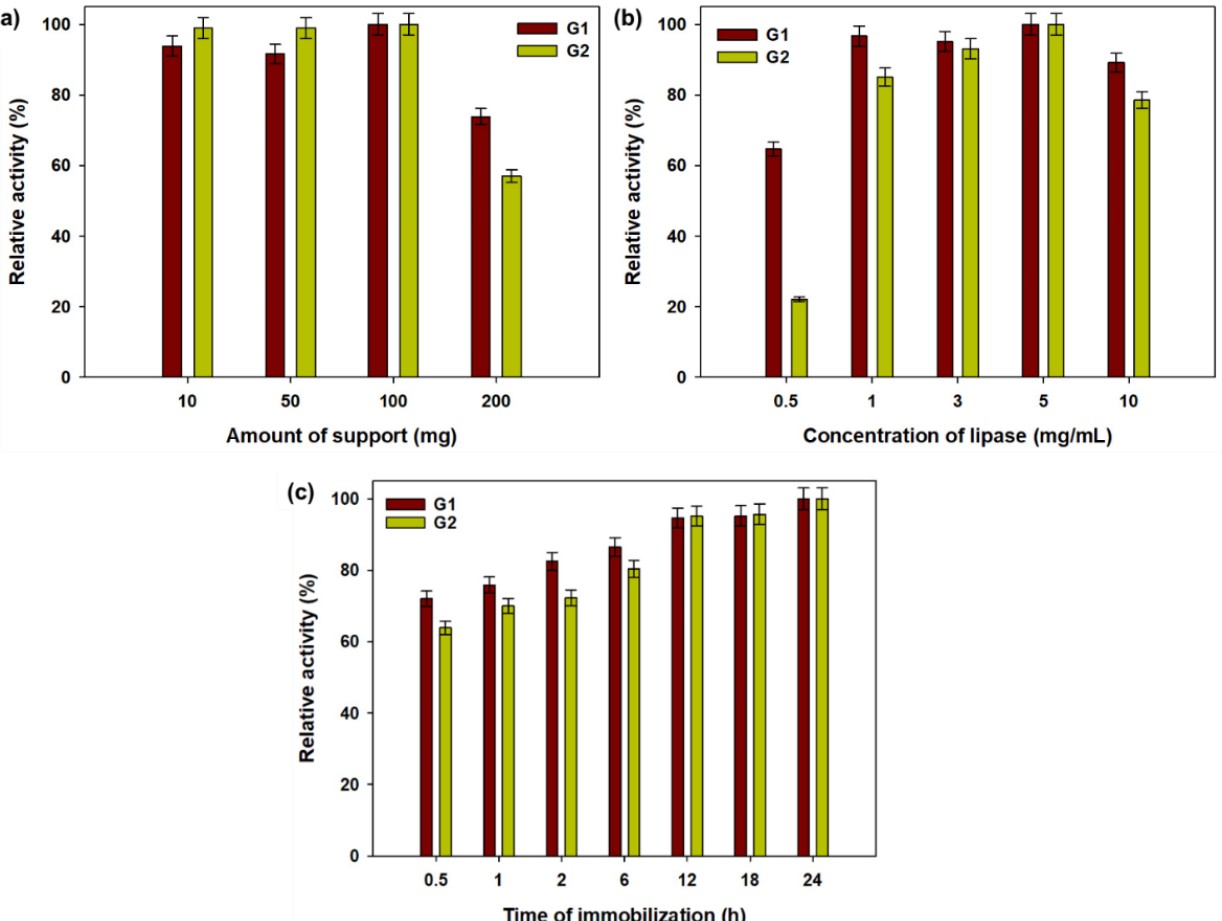

**Figure 5.** Relative activity of the immobilized enzymes depending on (**a**) amount of support, (**b**) concentration of protein, and (**c**) time of the immobilization process. All data are presented as the means ± standard deviation of three experiments.

Another factor to be considered is the time of the immobilization process, which may have an influence on the enzyme activity (see Figure 5c). The highest relative activity for lipase was observed after 24 h of immobilization, independently of the lignin-based spherical particles used. There is an almost linear correlation between the time of immobilization and the relative activity of the enzyme, which demonstrates a reduction in diffusional limitations in the transport of substrate to the attached enzyme [79–81]. However, the catalytic activity of lipase immobilized on material G1 is higher at every analyzed time point (especially in the range 0.5–6 h) and it increases with increasing process time, with the highest activity obtained after 24 h of immobilization. A duration of immobilization of 24 h was also suggested by Ameri et al., who prepared multiwalled carbon nanotubes and used them for adsorption of *Pseudomonas cepacia* lipase. It was observed that the optimum catalytic activity of the protein was highest (100%) after 24 h of the immobilization process [82]. Moreover, de França Serpa et al., who synthesized a novel magnetite–lignin hybrid material and used it as a support for the immobilization of lipase, reported that a 24 h time frame was the best for incubation of the protein [47]. Additionally, Lu et al. showed that after 24 h of lipase immobilization on nanoparticles, the enzyme exhibited high resistance to environmental conditions over a range of pH values from 3 to 11 [77].

### 2.3.1. Effect of pH and Temperature

The effect of pH on the relative activity of free and immobilized lipase was also analyzed, as free enzymes display high activity in narrow ranges of pH values (close to pH 7). Figure 6 shows the influence of pH, over a range from 5 to 9, on the catalytic activity

of the protein compared with that of the native lipase. The immobilized lipase exhibited maximum activity (100%) at pH 7, which is typical for this kind of enzyme [73,74]. Similar results were presented by Tomke et al., who also showed that free and immobilized lipase exhibited the highest cyclic activity at pH 6.8 [83]. Kamel Ariffin et al., who immobilized lipase from *Candida antarctica* on $Fe_2O_3$/chitosan particles, also stated that the relative protein activity was highest (100%) at pH 7 [84]. In another study, it was reported that only at this pH value (pH 7) the activity of lipase was over 90%. However, in more acidic or basic conditions, the activity of the protein was below 80%, and the results were similar independently of the support used [85]. It was also shown that the use of lignin-based spherical particles increases the activity of the immobilized enzyme (around 10%) over the entire tested range, compared with the native enzyme. It is suggested that the prepared immobilized systems cause an increase in stability and resistance against harsh environmental conditions, due to the stabilization of the enzyme structure upon immobilization, and also because the lignin spheres serve as protection for the enzyme molecules against external factors [86].

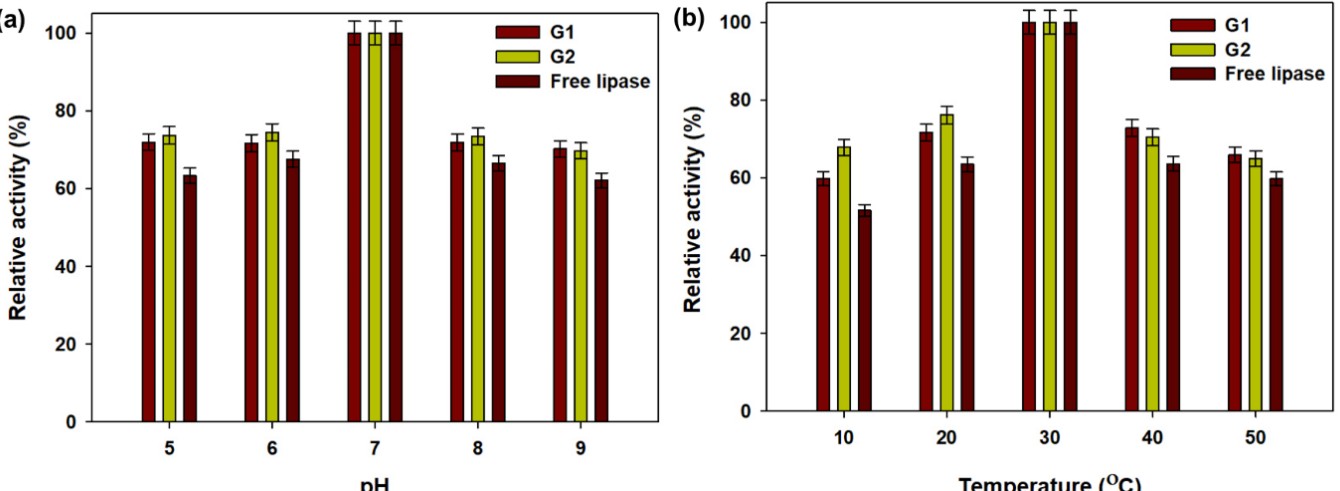

**Figure 6.** Effect of (**a**) pH and (**b**) temperature on the activity of native and immobilized lipase. All data are presented as the means ± standard deviation of three experiments.

Another important parameter affecting enzyme activity is temperature, which is an important variable determining the catalytic activity and cost of energy for industrial applications of immobilized systems. The hydrolytic activity of immobilized and native enzyme was determined in a temperature range from 10 to 50 °C, and the results are shown in Figure 6b. The highest activity of lipase was observed at 30 °C, independently of the support used, and with a change in temperature the activity of the protein decreased [75]. The highest relative activity at this temperature was found for the native enzyme extracted from *Candida antarctica*, which catalyzes the conversion most effectively under such neutral conditions [76]. Moreover, it should be noted that, in a lower temperature range (10 and 20 °C), the use of material G2 enables higher catalytic activity of the immobilized enzyme compared with material G1 and the free protein. At higher temperatures (40 and 50 °C) the use of material G1 is more favorable, which may be a consequence of the method of preparation of G1 at pH 5, as well as the smaller particle size of the support, the more developed structural properties, and the composition of functional groups. Over the entire range of tested conditions, the immobilized enzyme exhibited higher relative catalytic activity than the free enzyme; it can be concluded that the use of lignin-based spherical particles as a support can improve the temperature range suitable for application of the immobilized lipase. A similar observation was made by Qin et al., who stated that the activity of free and immobilized lipase on hydrogel spheres at 30 °C was at the level defined as 100%, and the activity decreased rapidly with an increase in temperature [87].

### 2.3.2. Thermal Stability

Catalytic reactions were performed after the free and immobilized enzyme had been stored for 3 h at 50, 60, and 70 °C. The results (Figure 7) show that the catalytic activity of lipase depends on the storage temperature. With an increase in the temperature from 50 °C to 70 °C, the activity of the free enzyme decreased from 62% to 40%. Immobilization of lipase on the surface of the supports increased its catalytic activity, especially with the use of material G2. The activity of the lipase immobilized on the G2 support remained the same (75%) at temperatures from 50 to 60 °C, and decreased slightly (to 62%) at 70 °C. These relatively good results may be connected to the enhanced protection of the immobilized enzyme by the support and its role as a heat capacitor. The largest difference between the catalytic activities of immobilized and free lipase was observed at 70 °C. At this temperature, the activity of lipase immobilized on lignin-based spherical particles was 58% and 62% for G1 and G2, respectively, whereas for free lipase the activity was around 20% lower.

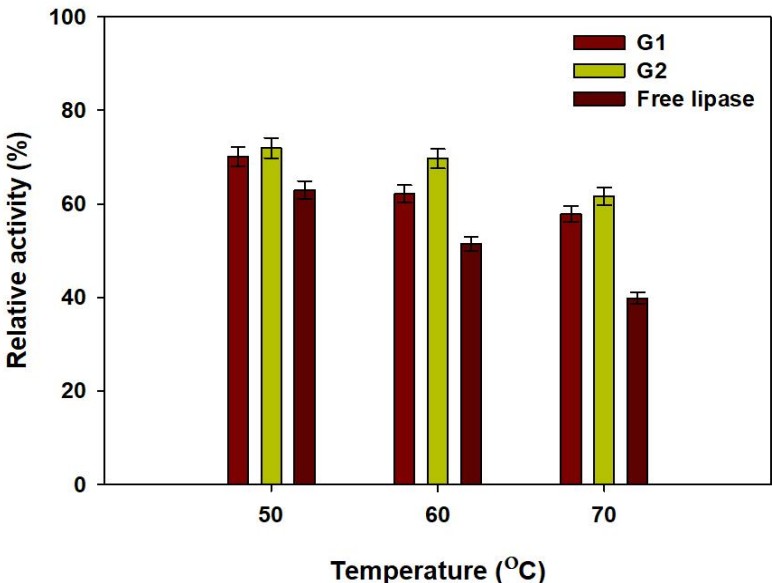

**Figure 7.** Thermal stability of free and immobilized lipase stored at 50, 60, and 70 °C for 3 h. A 100% relative activity corresponds to the results obtained at 30 °C. All data are presented as the means ± standard deviation of three experiments.

Pacheco et al. immobilized lipase on clay carriers, and tested the thermal stability of the systems at 45, 55, and 65 °C. Depending on the material used, the catalytic activity of the protein remained above 75% after 3 h of the procedure [88]. Similar results were presented by Lu et al., who immobilized lipase on $SiO_2$-coated $Fe_3O_4$ nanoparticles with the addition of polyacrylonitrile. In a temperature range from 50 to 60 °C, the immobilized enzyme retained catalytic activity above 50%, which was higher than for the free protein [77].

### 2.3.3. Reusability

For industrial applications, the reusability of the prepared systems is another significant factor. Both types of lignin-based spherical particles were tested over 10 catalytic cycles, and the results are shown in Figure 8. It is observed that the relative activity for both materials decreased over 10 catalytic cycles, and after the last cycle the relative activity was 87% and 55% for G1 and G2, respectively. The superior reusability of the biocatalytic system with material G1 possibly results from the fact that this material had better porous structure parameters, which may favor the deposition of the protein in the pores of the support as well as on its surface. This leads to better protection of lipase when this support is used, as well as limiting enzyme elution from the support. Consequently, the enzyme

may be better protected from the adverse effects of reaction conditions, and inactivation of the enzyme is limited.

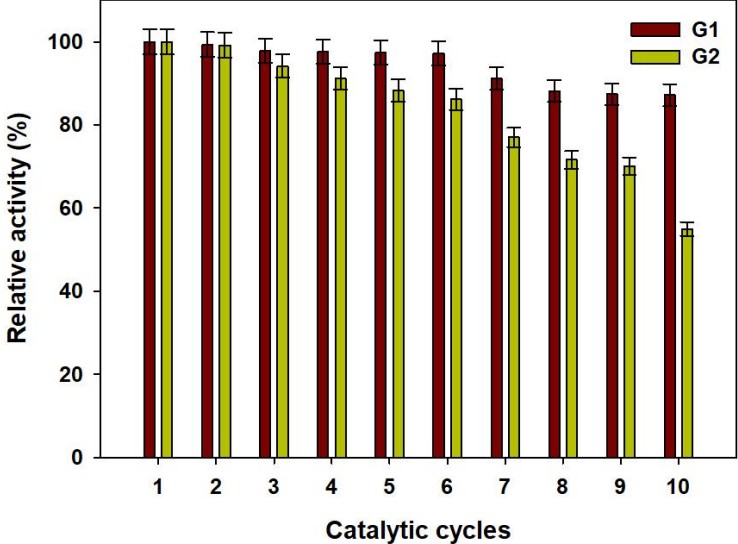

**Figure 8.** Reusability of lipase immobilized on lignin-based spherical particles. All data are presented as the means ± standard deviation of three experiments.

Qin et al. tested their hydrogel microspheres for potential reusability, and reported that after 10 cycles the relative activity of the enzyme was around 39% and 72% for materials on which 3 mg/mL and 20 mg/mL of lipase had been immobilized, respectively [87]. In turn, Tomke et al. immobilized lipase on coconut biochar and peanut shell residues. After more than five reuse cycles, the catalytic activity of the enzyme was measured at 58% [83]. Moreover, Cespugli et al. used rice husk for the immobilization of lipase, and after nine cycles its activity remained around 72% [89]. A similar experiment was performed by Costa-Silva et al., who also used rice husk for the immobilization of lipase, and after five cycles measured the catalytic activity at 81% [90]. Thus, it should be highlighted that lipase immobilized on both support materials can be reused with high efficiency, which is a great advantage compared with the use of free enzymes, which require additional and more complicated methods for protein recovery. In this case, the use of a solid support enables quick centrifugation of the solution and separation of the supernatant from sediment to retrieve the immobilized enzyme for further catalytic reaction. In summary, then, immobilization is highly favorable for the reusability of enzymes and for their potential applications.

## 3. Materials and Methods

### 3.1. Materials

Kraft lignin (MW ~10,000 kDa), (2-hydroxyethyl)trimethylammonium chloride (choline chloride), and ethyl alcohol, all purchased from Sigma Aldrich, Steinheim, Germany, were used for the preparation of lignin-based spherical particles. Sulfuric acid and sodium hydroxide solutions (Chempur, Piekary Śląskie, Poland) were used for adjustment of the pH during the synthesis of lignin-based material. The 50 mM acetate buffer, 50 mM phosphate buffer, 50 mM tris-buffer (all at specified pH), and sodium carbonate were supplied by Merck, Darmstadt, Germany. Lipase B from *Candida antarctica*—CALB (EC 3.1.1.3), p-nitrophenol, 15 mM p-nitrophenyl palmitate (p-NPP), and Bradford reagent were purchased from Sigma Aldrich, Steinheim, Germany.

### 3.2. Preparation of Spherical Matrix

Lignin was combined with (2-hydroxyethyl)trimethylammonium chloride for the preparation of spherical particles. Particles were synthesized in acidic, neutral, and basic conditions to verify the most suitable environmental conditions for the preparation of

lignin-based spherical materials. First, 0.1 g of choline chloride was dissolved in 10 mL of ethyl alcohol; then, 2 g of kraft lignin was also dispersed in ethyl alcohol in another flask. The lignin dispersion was combined with the choline chloride solution and mixed for 2 h. To obtain a clear solution, the lignin-based mixture was filtered, and then 500 mL of deionized water with the desired pH (3, 5, 7, 10) was added with the use of a peristaltic pump. The product was then filtered under vacuum with the use of a Sartorius Stedim system (Sartorius, Göttingen, Germany). Four different samples were prepared, as listed in Table 4 below.

**Table 4.** Symbols and pH conditions of the prepared materials.

| Sample | Reaction Conditions |
|--------|---------------------|
| G0 | pH 3 |
| G1 | pH 5 |
| G2 | pH 7 |
| G3 | pH 10 |

*3.3. Physicochemical Evaluation of Prepared Materials*

The surface morphology, shape, and size of the lignin-based particles prepared at different pH values were evaluated using scanning electron microscopy (SEM), obtained with a Mira-3 scanning electron microscope (Tescan, Brno, Czech Republic). The dispersive properties of the obtained materials, including particle size and polydispersity index, were measured with a Zetasizer Nano ZS apparatus (Malvern Instruments Ltd., Malvern, UK), which employs the non-invasive back scattering (NIBS) method to measure particle sizes ranging from 0.6 to 6000 nm. Prior to each measurement, a specific amount of the sample was sonicated in an ultrasonic bath (Polsonic, Warszawa, Poland) for 10 min to prepare a solid dispersion in a selected medium.

Characteristic functional groups present on the surface of the material before and after the immobilization process were identified with the use of Fourier transform infrared spectroscopy (FTIR). The analysis was performed over the wavenumber range 4000–450 $cm^{-1}$, with a resolution of 0.5 $cm^{-1}$, using a Vertex 70 apparatus (Bruker, Ettlingen, Germany). Approximately 2 mg of every sample was combined with 250 mg of potassium bromide (KBr), placed in a steel ring under 10 MPa, to form a tablet. Attenuated total reflectance (ATR) spectroscopy was used to obtain the spectrum for choline chloride. The analysis was performed using a single-reflection diamond ATR accessory (Platinum ATR, Bruker Optics GmbH, Ettlingen, Germany).

A Zetasizer Nano ZS instrument (Malvern Instruments Ltd., Malvern, UK) was also employed to measure the electrophoretic mobility of the lignin-based spherical particles before and after the immobilization process. The instrument was equipped with an autotitrator (Malvern Instruments Ltd., Malvern, UK). The zeta potential was calculated, and electrokinetic curves were plotted. The measurements were carried out at pH values ranging from 2 to 10, and the pH was adjusted with the use of 0.2 mol/L hydrochloric acid and 0.2 mol/L sodium hydroxide. In total, 10 mg of the tested material was sonicated in an ultrasonic bath (Polsonic, Warszawa, Poland) to form a stable dispersion in 0.001 mol/L sodium chloride electrolyte.

The porous structure properties, including the BET surface area ($A_{BET}$), total volume of pores ($V_p$), and mean size of pores ($S_p$), were obtained with the use of an ASAP 2020 apparatus (Micrometrics Instrument Co., Norcross, GA, USA). The surface area was determined by the Brunauer–Emmet–Teller method, and the total volume and mean size of pores were evaluated using the Barret–Joyner–Halenda (BJH) algorithm. All samples were degassed prior to measurement.

*3.4. Lipase Immobilization*

The examined parameters of the immobilization process included the initial amount of the support, the process duration, and the concentration of the enzyme solution. For this

purpose, from 10 to 200 mg of the previously obtained G1 and G2 material, which exhibited a spherical shape and high homogeneity, were placed in vials, and 5 mL of enzyme solution at concentrations of 0.5, 1, 3, 5, and 10 mg/mL was added. The immobilization process was carried out in phosphate buffer at pH 7 for 0.5, 1, 2, 6, 12, 18, and 24 h in an Eppendorf Thermomixer C (Eppendorf, Hamburg, Germany) at a temperature of 30 °C. After the process, the obtained products were centrifuged (Eppendorf Centrifuge 5810 R, Eppendorf, Hamburg, Germany) at 4000 rpm, filtered, washed several times to remove the unbound enzyme, and dried at ambient temperature for 24 h. All immobilization processes were performed in each condition in triplicate and the results are presented as the mean value from three experiments.

*3.5. Hydrolytic Activity and Immobilization Characterization*

The relative hydrolytic activity of the free and immobilized lipase was determined in the model hydrolysis reaction of p-nitrophenyl palmitate (p-NPP) to p-nitrophenol (p-NP) and palmitic acid. All reactions were performed in triplicate and were carried out with stirring at 300 rpm at 30 °C for 5 min in 3 mL of a buffer solution at the desired pH and molarity. The reactions with the free enzyme were performed in the same conditions as for the immobilized lipase. The amount of remaining p-nitrophenyl palmitate was measured spectrophotometrically at $\lambda = 300$ nm using a Jasco V-750 UV-Vis spectrophotometer (Jasco, Tokyo, Japan). A standard calibration curve for p-nitrophenyl palmitate was plotted to calculate the relative activity of the immobilized lipase. The highest activity obtained was defined as 100% relative activity. Further, using the standard calibration curve for p-NPP, the specific activity of the free and immobilized enzyme (U/mg) was calculated, and thus the activity recovery (%) of the immobilized lipase was presented as the percentage activity of the immobilized enzyme relative to the catalytic activity of the free enzyme.

The Bradford method was used for calculation of the amount of immobilized enzyme. The measurements were taken spectrophotometrically at wavelength 595 nm by analyzing the supernatant after immobilization. Bradford reagent was mixed with the analyzed lipase solution in a 1:1 ratio, and the analysis was carried out after 10 min. Equation (1) was used to calculate the activity recovery of the immobilized lipase:

$$\text{Activity recovery } (\%) = \frac{A_I}{A_0} \cdot 100\% \tag{1}$$

Moreover, the immobilization yield (%) was calculated according to Equation (2):

$$\text{Immobilization yield } (\%) = \frac{A_0 - A_f}{A_0} \cdot 100\% \tag{2}$$

where $A_0$ is the total initial activity of the free lipase, $A_I$ is the activity of the immobilized enzyme, and $A_f$ is the total activity of the enzyme in the supernatant and washing solution after immobilization.

The difference between the initial amount of lipase and the final amount of the enzyme in the supernatant after immobilization was used to determine the amount of enzyme immobilized, relative to the mass of the support. To determine the effect of process conditions on activity of immobilized enzyme, biocatalytic systems obtained at following conditions were used: 30 °C, pH 7, enzyme concentration 5 mg/mL, 100 mg of lignin support, and a 24 h process duration.

3.5.1. Effect of pH

The hydrolysis reaction of *p*-NPP to *p*-NP was carried out at 30 °C for 5 min in 3 mL of a specific buffer, under constant stirring, to evaluate the effect of pH. The reaction was performed for native and immobilized lipase at different pH values, ranging from 5 to 9. The activity of the remaining substrate was measured spectrophotometrically according to the methodology presented above.

### 3.5.2. Effect of Temperature

The effect of temperature on lipase activity was established after the hydrolysis reaction of *p*-NPP to *p*-NP catalyzed by the free or immobilized enzyme at temperatures varying from 10 to 50 °C. The reaction was carried out for 5 min at pH 7 in 3 mL of 50 mM phosphate buffer, with constant stirring. After the process, the samples were measured spectrophotometrically.

### 3.5.3. Temperature Stability

The model hydrolysis reaction of *p*-nitrophenol palmitate was used to determine the thermal stability of the native lipase and the product following immobilization. The hydrolytic activity was examined after 3 h incubation of the produced systems in 3 mL of 50 mM phosphate buffer at pH 7 at 50, 60 and 70 °C.

### 3.5.4. Reusability

Reusability was evaluated based on the model hydrolysis reaction presented above. After every hydrolysis step, the immobilized material was centrifuged to separate the reaction mixture. The particles were washed with phosphate buffer (pH 7), dried at ambient temperature, and used for another reaction cycle (up to 10 repetitions).

### *3.6. Statistical Analysis*

All measurements were made in triplicate. Error bars are presented as the mean $\pm$ standard deviation of three experiments. Statistically significant differences were determined using Tukey's test by one-way ANOVA, performed in SigmaPlot 12.0 (Systat Software Inc., Chicago, IL, USA). Statistical significance was established at the level $p < 0.05$.

### 4. Conclusions

In this study, lignin-based spherical particles were formed with the use of choline chloride at different pH values and were used as support for the immobilization of lipase. The main goal was to prepare lignin-based spherical particles and to verify which reaction conditions would be the most favorable for the preparation of samples suitable for enzyme immobilization. It was found that the smallest spherical particles were obtained in mildly acidic conditions (pH 5), but the preparation of samples at pH 7 also resulted in a lignin material with particles of spherical shape. Compared with pristine kraft lignin, both materials exhibited improved porous structure parameters. In the next step, samples G1 and G2 were used as support for the immobilization of lipase, and it was confirmed that both materials show potential for the adsorption of enzymes. The most suitable conditions for this procedure were found to be 24 h of incubation, a temperature of 30 °C, pH 7, and the use of 100 mg of support with 5 mg/mL initial enzyme concentration. It was also found that lipase immobilized on material G1 exhibited higher catalytic activity after immobilization over a wide range of process conditions, including pH and temperature. Material G2 proved more suitable as a support at higher pH values and temperature ranges; therefore, it improves the protein's thermal stability. However, lipase immobilized on material G1 retains approximately 87% of its catalytic activity after 10 repeated reaction cycles, which is important for practical application possibilities. Finally, it appears that the G1 support is prone to stabilize the enzyme structure during the immobilization process. Therefore, the proposed method of preparation of lignin-based spherical particles with the use of choline chloride may have potential application as a means of fabricating supports for the immobilization of enzymes for various industrial uses.

**Author Contributions:** Conceptualization, M.S. and T.J.; methodology, M.S., K.B., Ł.K., J.Z. and T.J.; software, M.S., K.B., K.S.-C., Ł.K. and J.Z.; validation, M.S., K.B., Ł.K. and J.Z.; formal analysis, M.S., K.B., Ł.K. and J.Z.; investigation, M.S., K.B., Ł.K., K.S.-C. and J.Z.; resources, M.S., K.B., Ł.K. and J.Z.; data curation, M.S., K.B., K.S.-C., Ł.K. and J.Z.; writing—original draft preparation, M.S., K.B.; writing—review and editing, M.S., K.B., Ł.K., J.Z. and T.J.; visualization, M.S., K.B., Ł.K. and J.Z.;

supervision, T.J.; project administration, T.J.; funding acquisition, T.J. All authors have read and agreed to the published version of the manuscript.

**Funding:** This research was funded and prepared as part of a research project supported by the National Science Center Poland, no. 2017/27/B/ST8/01506.

**Data Availability Statement:** Not applicable.

**Conflicts of Interest:** The authors declare no conflict of interest.

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
