# Peer review of "Tailoring Lignin-Based Spherical Particles as a Support for Lipase Immobilization"

_catalysts, doi:10.3390/catal12091031_

Round 1

Reviewer 1 Report

    While the overall scientific merits of the manuscript are sound, additional critical details in the methods and results are necessary.

MATERIALS AND METHODS

-          3.4

o   Line 524.  Indicate if the enzyme concentration is on a protein basis

o   Is the CALB purified prior to use?

o   Indicate if the immobilization process was reproduced for each condition or done only once.

-          3.5

o   More details on the activity assay are needed.  Include the reaction volume, enzyme concentration (or range of concentrations), buffer concentration and pH, pNPP concentration, etc.

o   Line 533.  Indicate if “triplicate” is technical or biological replicates.

o   Indicate if the Bradford method was used to indirectly measure the immobilized enzyme concentration (i.e., the amount of protein in the supernatant after immobilization) or directly on the particles.

o   Indicate if naked particles (i.e., particles without offered protein) were used as a control for the Bradford and enzyme activity assays (particles and leached components of particles can sometimes interfere with these assays).

o   Equation 1: Is “Catalytic Activity” the same as “Activity recovery” in Equation 3?  If so, be consistent with the terms.

o   Equation 2: Immobilization yield (or immobilization efficiency) is typically based on protein concentration.  Equation 2 appears to be based on activity.  Since activity can increase or decrease with immobilization, protein concentration should be used to determine immobilization yield. 

o   Indicate the specific immobilization condition and the number of biological replicates used to evaluate the effect of pH, temperature, temperature stability, and reusability.

RESULTS

-          For all results with error bars, indicate if the error bars are standard deviation or standard error.  Also, indicate sample size (n) for data in the figures.

-          2.3

o   Table 3.  Indicate the range or standard deviation associated with the results.  Also, indicate the conditions for immobilization in the title.

o   Line 319:  Why was this reference singled out?  CALB has been immobilized by hydrophobic adsorption to several supports.  Please indicate how this work compares to other supports in terms of optimum hydrolytic activity.

o   Line 322: Indicate the p-value used for significance when stating, “A significant decrease in protein activity….”

o   Lines 324 – 326:  Please rewrite the sentence beginning with “Although in the case of….” for clarity.

o   Line 326-329:  Citation is needed to support the statement “This is due to the fact….”.  It is not uncommon for immobilized enzymes at low enzyme/unit area concentrations to have reduced activity due to limited crowding, which allows the enzyme to “spread” on the surface.  Too much enzyme can lead to multiple layers or blocked active sites.  Often there is an optimum.

o   Lines 377 – 382.  It is unclear if these statements (“It was also reported….”) are based on the cited work or the work presented in Figure 6.  Is the 10% increase in activity in terms of absolute or relative activity?  Is the difference significant? 

CONCLUSION:

-          Indicate in which ways the developed process is advantageous (if any) to current commercial approaches for CALB immobilization  

Author Response

Dear Reviewer,

Thank you for your insightful review of our work, which contributed to a better understanding of the scientific problems related to the subject of the publication and will help with the elimination of potential errors in the future.We would also like to express our gratitude for the revision of our manuscript and the opportunity to re-submit it, incorporating all of the Referees’ suggestions. Our comments and changes are noted below and are marked in yellow in the manuscript. 

Response to Reviewer #1:

While the overall scientific merits of the manuscript are sound, additional critical details in the methods and results are necessary. 

Comment 1: Line 524.  Indicate if the enzyme concentration is on a protein basis.

Answer 1: We thank the Reviewer for this remark. In our research, a commercial CALB in solid form was used to obtain a solution by dissolving the enzyme in a suitable buffer solution. Initially, the solution with the highest concentration of the enzyme needed in studies (10 mg/mL) was prepared as a stick solution, and then a series of dilutions was made to obtain the other concentrations (0.5, 1, 3 and 5 mg/mL). 

Comment 2: Is the CALB purified prior to use?

Answer 2: We would like to explain that in our research, commercial CALB was used, thus the enzyme was ready for use and it was not necessary for further purification prior to immobilization and biocatalytic processes. 

Comment 3: Indicate if the immobilization process was reproduced for each condition or done only once.

Answer 3: We thank the Reviewer for this remark. Immobilization was carried out in triplicate in each of the mentioned process conditions. The changes have been introduced in the section 3.4, as below:“All immobilization processes were performed in each conditions in triplicate and results are presented as a mean values from three experiments.”

Comment 4: More details on the activity assay are needed. Include the reaction volume, enzyme concentration (or range of concentrations), buffer concentration and pH, pNPP concentration, etc.

Answer 4: We thank the Reviewer for this suggestion. We have added the reaction parameters into the different subchapters of Materials and Methods section in the revised manuscript. The changes have been marked in yellow.

Comment 5: Line 533.  Indicate if “triplicate” is technical or biological replicates.

Answer 5: We thank the Reviewer for this remark. In our case, "triplicate" means that the immobilization or hydrolysis reactions were repeated three times under each of the mentioned conditions. Thus, each sample was analyzed once, but due to the three repetitions, finally, for each condition, three results were obtained, from which mean value and error bars could be calculated.  

Comment 6: Indicate if the Bradford method was used to indirectly measure the immobilized enzyme concentration (i.e., the amount of protein in the supernatant after immobilization) or directly on the particles.

Answer 6: We thank the Reviewer for this comment. After completion of immobilization process, the sample was centrifuged, and then the supernatant was separated from the support with protein. Subsequently, the supernatant was mixed with the Bradford reagent and it was possible to check the amount of enzyme that did not deposit to the support. Finally, based on the knowledge of the initial amount of protein, it was possible to estimate the amount of the immobilized enzyme. Thus, in our studies we used indirect measurements. Proper explanations have been provided in section 3.5 of the revised manuscript and marked inyellow. 

Comment 7: Indicate if naked particles (i.e., particles without offered protein) were used as a control for the Bradford and enzyme activity assays (particles and leached components of particles can sometimes interfere with these assays).

Answer 7: We thank the Reviewer for this suggestion. We fully agree with this comment and we also realize that spectrophotometric measurements can be easily disturbed by the presence of particles and we should be careful with the Bradford method or enzyme activity assays. Therefore the reference sample has been prepared to obtain a control sample for all of the measurements. For this reason, pristine particles of support in an appropriate buffer solution have been shaken and after appropriate the sample was centrifuged, and then the obtained supernatant was used as a control sample during the measurements. 

Comment 8: Equation 1: Is “Catalytic Activity” the same as “Activity recovery” in Equation 3?  If so, be consistent with the terms.

Answer 8: We thank the Reviewer for this comment. It has to be emphasized that "Catalytic activity" and "Activity recovery" are not the same parameters. Due to the fact that "Activity recovery" is calculated based on the activity of the immobilized enzyme and the activity of the free enzyme. Moreover, in our studies, "Activity recovery" was estimated to characterize the differences between the biocatalytic system before and after enzyme immobilization. While the "Catalytic activity" values were used to determine the efficiency of the p-NPP hydrolysis and activity of native or immobilized CALB under various process conditions. 

Comment 9: Equation 2: Immobilization yield (or immobilization efficiency) is typically based on protein concentration. Equation 2 appears to be based on activity. Since activity can increase or decrease with immobilization, protein concentration should be used to determine immobilization yield.

Answer 9: We thank the Reviewer for this remark. We fully agree that the immobilization efficiency can be calculated with the use of the enzyme concentration. However, immobilization yield is usually calculated based on enzyme activity. Moreover, we are aware that there is a debate about the validity of calculating the immobilization yield using protein activity. Nevertheless, in many publications this parameter is estimated in this way, therefore, in our research, such calculations seem justified, so that it could be possible to compare with the results in other articles. This issue has been described in an interesting way and with more detail in https://doi.org/10.1016/j.procbio.2019.11.026.  

Comment 10: Indicate the specific immobilization condition and the number of biological replicates used to evaluate the effect of pH, temperature, temperature stability, and reusability.

Answer 10: We thank the Reviewer for this comment. We have added the volume of the reaction in sections from 3.5.1 to 3.5.3. The number of biological replicates was added into the 3.6. section which is related to all performed effects of pH, temperature, temperature stability and reusability. In section 3.5. we have also specified the immobilization conditions used to prepare systems for further analysis. Following sentence has been added: “To determine the effect of process conditions on activity of immobilized enzyme, biocatalytic systems obtained at following conditions were used: 30 °C, pH 7, enzyme concetration 5 mg/mL, 100 mg of lignin support and 24 h of process duration.”

Comment 11: For all results with error bars, indicate if the error bars are standard deviation or standard error.  Also, indicate sample size (n) for data in the figures.

Answer 11: We thank the Reviewer for this suggestion. In our studies the error bars are standard deviation, therefore in the current version of manuscript the following explanation has been added to figure captions: “All data are presented as means ± standard deviation of three experiments.”

Comment 12: Table 3. Indicate the range or standard deviation associated with the results.  Also, indicate the conditions for immobilization in the title.

Answer 12: We thank the Reviewer for this comment. The values of standard deviation were introduced to Table 3, and the process conditions under which the enzyme immobilization were carried out were added to the caption. 

Comment 13: Line 319: Why was this reference singled out? CALB has been immobilized by hydrophobic adsorption to several supports. Please indicate how this work compares to other supports in terms of optimum hydrolytic activity.

Answer 13: We thank the Reviewer for this comment. We have singled this reference out, because the research also stated that 100 mg of a support is optimal for conducting of immobilization. Moreover, like in our studies, the proposed support material was also in spherical shape and involved biopolymers. Additionally, it should be highlighted, the usage of lignin-based spherical particles is relatively new and therefore the data is still limited. Hence, we wanted to compare obtained data with results on lipase immobilization on biopolymers. 

Comment 14: Line 322: Indicate the p-value used for significance when stating, “A significant decrease in protein activity….”

Answer 14: We thank the Reviewer for this suggestion. In order to complete the information related to statistical analysis and the number of repetitions of processes, a new Section 3.6 has been added to the Materials and Methods, as below:“3.6. Statistical analysisAll measurements were made in triplicate. Error bars are presented as means ± standard deviation. Statistically significant differences were determined using Tukey’s test by one-way ANOVA performed in SigmaPlot 12 (Systat Software Inc., USA). Statistical significance was established at the level p < 0.05.” 

Comment 15: Lines 324–326: Please rewrite the sentence beginning with “Although in the case of….” for clarity.

Answer 15: We thank the Reviewer for this remark. The sentence has been rewritten, as below:“In the case of G1 the enzyme activity decreased with the use of a smaller amount of support. However, when 200 mg was used for G1 material, the activity of lipase was higher than with material G2.” 

Comment 16: Line 326–329: Citation is needed to support the statement “This is due to the fact….”. It is not uncommon for immobilized enzymes at low enzyme/unit area concentrations to have reduced activity due to limited crowding, which allows the enzyme to “spread” on the surface.  Too much enzyme can lead to multiple layers or blocked active sites.  Often there is an optimum.

Answer 16: We would like to thank the Reviewer for this suggestion. We fully agree with the Reviewer and therefore we have added the proper reference to the manuscript.  

Comment 17: Lines 377–382: It is unclear if these statements (“It was also reported….”) are based on the cited work or the work presented in Figure 6.  Is the 10% increase in activity in terms of absolute or relative activity?  Is the difference significant?

Answer 17: We would like to thank the Reviewer for this suggestion. We have rephrased this statement to make it clearer and to avoid misunderstandings. The difference (10% increase in relative activity) is quite significant because is not between activity values in different pH, but between the relative activities of native and immobilized lipase in the same pH value. Moreover, this difference in activities may seem insignificant in one biocatalytic cycle but presented immobilized material can be used several times therefore it is important for further applications possibilities. 

Comment 18: Indicate in which ways the developed process is advantageous (if any) to current commercial approaches for CALB immobilization 

Answer 18: We would like to thank the Reviewer for this comment. We have prepared lignin-based spherical particles with the use of choline chloride, which to our best knowledge has never been performed before. It is a novel support, which was prepared with the use of eco-friendly materials. Another advantages is the possibility to use a lignin that act as a waste material in a numerous of processes including mainly paper and wood industry that makes produced materials relatively cheap and easy to prepare. Further, proposed support possess a numerous of functional groups capable for stable enzyme binding that enhances activity of the produced materials and limits enzyme leaching. The immobilized enzyme shows good reusability, which could be used in the future for commercial approaches. The results that were obtained so far allow us to think optimistically and to state that the next step in our research will be to scale-up of the processes we conduct. 

The whole manuscript has also been carefully checked with regard to editorial and language issues.

We look forward to hearing from you.

Yours faithfully,

Professor Teofil Jesionowski

corresponding author

Reviewer 2 Report

This work is of interest for the development of practical applications of biocatalytic processes in various directions. Increasing enzyme cycles plays an important role in terms of the economics of such technologies. Well conducted and written study. But there are some minor comments:

1. At the beginning of abstract it is said that the advantage of using lignin-based particles is the possibility of simultaneous disposal of utilizing industrial waste for their preparation. However, the lignin that is used in the work is not a waste and has been successfully used in the recovery of chemicals and energy after wood pulping. I recommend correcting the first sentence in this regard.

2. Although choline chloride is one of the keywords in the work, they are missing from the abstract. It's good to add this.

3. Kraft lignin was used in the work, but its characteristics are not: from softwood or hardwood, molecular weight distribution, etc.

4. The pH value of the medium during the analysis of the activity of free and immobilized lipase is not indicated.

After correcting small but important remarks, the article is recommended for publication in the journal.

Author Response

Dear Reviewer, 

Thank you for your insightful review of our work, which contributed to a better understanding of the scientific problems related to the subject of the publication and will help with the elimination of potential errors in the future.We would also like to express our gratitude for the revision of our manuscript and the opportunity to re-submit it, incorporating all of the Referees’ suggestions. Our comments and changes are noted below, and are marked in yellow in the manuscript. 

Response to Reviewer #2: This work is of interest for the development of practical applications of biocatalytic processes in various directions. Increasing enzyme cycles plays an important role in terms of the economics of such technologies. Well conducted and written study. But there are some minor comments: 

Comment 1: At the beginning of abstract it is said that the advantage of using lignin-based particles is the possibility of simultaneous disposal of utilizing industrial waste for their preparation. However, the lignin that is used in the work is not a waste and has been successfully used in the recovery of chemicals and energy after wood pulping. I recommend correcting the first sentence in this regard.

Answer 1: We would like to thank the Reviewer for this suggestion. We have changed the sentence as follows: “Lignin-based spherical particles have recently gained popularity due to their characteristic and the usage of biopolymeric material.” 

Comment 2: Although choline chloride is one of the keywords in the work, they are missing from the abstract. It's good to add this.

Answer 2: We would like to thank the Reviewer for this comment. We have added the sentence “choline chloride” to the abstract.  

Comment 3: Kraft lignin was used in the work, but its characteristics are not: from softwood or hardwood, molecular weight distribution, etc.

Answer 3: We would like to thank the Reviewer for this comment. We have bought this material in Sigma-Aldrich and it is not stated which type of wood was used for obtaining of this type of lignin, however we have added into the manuscript the molecular weight of the lignin in Materials and Methods section.  

Comment 4: The pH value of the medium during the analysis of the activity of free and immobilized lipase is not indicated.

Answer 4: We thank the Reviewer for this comment. The process conditions (including the pH of the medium) in which the characterization of the biocatalytic system with free or immobilized enzyme was carried out, have been added to the caption of Table 3. 

The whole manuscript has also been carefully checked with regard to editorial and language issues.

We look forward to hearing from you.

Yours faithfully,

Teofil Jesionowski

corresponding author

Round 2

Reviewer 1 Report

Questions and comments have been adequately addressed, and the manuscript has been sufficiently revised.